# Four Social Brain Regions, Their Dysfunctions, and Sequelae, Extensively Explain Autism Spectrum Disorder Symptomatology

**DOI:** 10.3390/brainsci9060130

**Published:** 2019-06-04

**Authors:** Charles S. E. Weston

**Affiliations:** Independent researcher, 4 rue des Sablonnières, 37400 Amboise, France; weston_charles@hotmail.com; Tel.: +33-9-8359-7630

**Keywords:** autism, pathogenic mechanisms, amygdala, orbitofrontal cortex, temporoparietal cortex, insula, intangible knowledge, paradoxical functional facilitation, biomarker

## Abstract

Autism spectrum disorder (ASD) is a challenging neurodevelopmental disorder with symptoms in social, language, sensory, motor, cognitive, emotional, repetitive behavior, and self-sufficient living domains. The important research question examined is the elucidation of the pathogenic neurocircuitry that underlies ASD symptomatology in all its richness and heterogeneity. The presented model builds on earlier social brain research, and hypothesizes that four social brain regions largely drive ASD symptomatology: amygdala, orbitofrontal cortex (OFC), temporoparietal cortex (TPC), and insula. The amygdala’s contributions to ASD largely derive from its major involvement in fine-grained intangible knowledge representations and high-level guidance of gaze. In addition, disrupted brain regions can drive disturbance of strongly interconnected brain regions to produce further symptoms. These and related effects are proposed to underlie abnormalities of the visual cortex, inferior frontal gyrus (IFG), caudate nucleus, and hippocampus as well as associated symptoms. The model is supported by neuroimaging, neuropsychological, neuroanatomical, cellular, physiological, and behavioral evidence. Collectively, the model proposes a novel, parsimonious, and empirically testable account of the pathogenic neurocircuitry of ASD, an extensive account of its symptomatology, a novel physiological biomarker with potential for earlier diagnosis, and novel experiments to further elucidate the mechanisms of brain abnormalities and symptomatology in ASD.

## 1. Introduction

Autism spectrum disorder (ASD) is a familiar neurodevelopmental disorder, with a complex and heterogeneous symptomatology that normally persists throughout life. Estimates of the prevalence of ASD have increased over the years, largely for methodological reasons but likely for others as well [1,2,3,4]. Recent prevalence estimates of ASD range from 1.46–2.50% [5,6,7]. ASD is markedly heterogeneous, and some researchers regard it as a family of disorders, “the autisms” [8]. For instance, language function may range from absence of any language to largely typical levels of competence [9,10]. Again, IQ may range from intellectual disability (ID), found in about two thirds of ASD individuals, through average IQ, to high IQ [11,12,13].

Knowledge of ASD symptomatology continues to develop, and it has become evident that ASD symptomatology encompasses multiple abnormalities in the social, language, sensory, motor, cognitive, emotional, repetitive behavior, self-care, and daily living skills domains, and many of these often start to emerge during infancy [14,15,16,17,18,19,20,21,22,23,24,25,26,27]. ASD is often a distressing, unhappy state for the sufferer [22]. For their families and carers, it is enormously stressful and troubling, and detrimental to many aspects of life including aspirations, marriage, finances, wellbeing, and health [28,29,30,31,32,33]. Early diagnosis and commencement of treatment, however, yield clinical benefits and markedly reduced care costs [7,34,35]. 

Together, ASD is a complex, heterogeneous, disabling, and refractory disorder. Importantly, ASD seems to be a unitary disorder. Constantino et al. examined the factor structure of ASD by applying several statistical procedures, cluster analysis, and principal components factor analysis to data collected by standard questionnaires from parents and teachers of ASD children and adolescents. The findings were that there was a single unitary factor underlying ASD symptomatology [36]. Such findings have been replicated by other research groups [37,38]. Thus, there is likely a common pathogenic mechanism that underlies ASD, with heterogeneity of symptomatology largely deriving from variations in severity. 

In this work, the research question examined is: What are the pathogenic neural circuits that explain ASD symptomatology in all its richness? The major symptoms and features of ASD are first summarized, as these objectively set out the scientific challenge, and what has to be explained by a model of the pathogenic mechanisms. A theoretical model of ASD is then presented that sets out the major neurocircuitry disruptions of the disorder, which, through direct and indirect dysfunctions, can extensively explain ASD symptomatology and features. 

At the level of disrupted neurocircuitry, disruption of four social brain regions is hypothesized to largely drive ASD symptomatology, and these social brain regions are the amygdala, orbitofrontal cortex (OFC), temporoparietal cortex (TPC), and insula. The model builds on the rich body of research and hypotheses of social brain region involvement in ASD (e.g., [39,40,41,42,43,44]). Further brain regions commonly display structural and functional abnormalities in ASD, including visual cortical areas, prefrontal cortex (PFC) subregions, caudate nucleus, and putamen subregions of the basal ganglia, hippocampus, sensorimotor cortex, cerebellum, and thalamus [45,46,47,48,49]. A number of these abnormalities, however, are hypothesized to be driven by the four disrupted social brain regions or by ASD symptoms, and relevant evidence is summarized. That is, such abnormalities are interpreted in terms of the established concept of the spread of disturbance to strongly interconnected brain regions or related concepts [50,51,52,53]. 

At the symptomatology level, many symptoms are hypothesized to flow directly from the four social brain regions’ disruptions, and are consistent with those brain regions’ known processing. This information is set out, and is intended to comply with the framework of the research domain criteria (RDoC), in that the disordered neurocircuitry level of analysis is related to disturbed features, behaviors, cognitions, and other symptom domains [54,55,56]. In addition, many further symptoms are sequelae or consequences of the directly caused disruptions and dysfunctions, and these are summarized together with supporting empirical evidence. That is, such symptoms are interpreted in terms of disruption of strongly interconnected brain regions, Wing’s “secondary behaviour problems”, compensatory, and adaptive responses [50,51,52,53]. For simplicity, these are collectively termed sequelae. Together, substantive explanations are offered for extensive ASD symptoms and features including: disordered visual scanpaths, apparent social disinterest, indiscriminate visual and auditory processing, abnormal sensory sensitivities, preference for concrete-level cognition, impaired conceptual processing, rote learning, inflexible repetitive routines, aspects of poor self-care and impaired daily living skills, stress, dysphoric emotions, stereotypies, and self-injurious behaviors (SIBs). Thus, the model has substantial explanatory power. In addition, clarification of those symptoms and features that are directly caused and those that are sequelae is novel and important as it brings some order to the complex symptomatology of ASD, and it should constrain and focus the search for the pathogenic mechanisms of ASD. Further, the model should facilitate the development of interventions that are based on pathogenic mechanisms.

## 2. Major Symptoms and Features of ASD

ASD comprises a multiplicity of abnormalities, commonly affecting sensory, motor, cognitive, emotional, repetitive behavior, activities of daily living, social, and language domains [26,57,58,59,60], and these are summarized in turn below. The essence of ASD symptomatology is summarized by Jolliffe et al. [61]. 

“I feel it [the definition of autism] should have read, ‘an inability to understand reality in the first place and that this itself leads to a person being withdrawn’. This is where the problem in forming relationships and relating to other people comes in. The latter is difficult to do because you have never been able to make any sense of reality, and thus cannot understand why you should, and how you should, form a relationship, not that you have just withdrawn from reality. Reality to an autistic person is a confusing, interacting mass of events, people, places, sounds and sights. There seem to be no clear boundaries, order or meaning to anything. A large part of my life is spent just trying to work out the pattern behind everything. Set routines, times, particular routes and rituals all help to get order into an unbearably chaotic life……It is the confusion that results from not being able to understand the world around me which I think causes all the fear. This fear then brings a need to withdraw.” (pp. 15–16). 

These observations are supported by additional accounts written by high-functioning ASD individuals or those closest to them (e.g., [62,63,64,65]). ASD symptomatology is being further elucidated by empirical investigations, as summarized next.

### 2.1. Sensory Abnormalities 

Low-level visual functions were found to be essentially intact in an extensive review of the empirical literature [66], but high-level visual abnormalities are commonly reported. A number of experimental studies have recorded with eye-tracking instruments the visual scanpaths that are executed during the viewing of social scene pictures or video clips by ASD and typically developing (TD) subjects. The findings are that TD subjects execute organized scanpaths that are commonly directed to meaningful and significant components of scenes, such as human faces, eyes, and mouths, and the objects with which they interact. In contrast, ASD subjects execute somewhat disorganized and atypical scanpaths through scenes, that include fixations on meaningless areas, such as parts of surfaces, or of the background, or near peripheral objects such as a light switch ([67,68,69,70,71,72,73,74,75]; see for ASD studies meta-analysis, [76]). 

A further common visual abnormality is that ASD individuals frequently fail to filter visual stimulation. TD individuals visually process only limited components of the visual environment, mainly those that are salient or meaningful in some way (see preceding paragraph [68]). In contrast, ASD individuals visually process indiscriminately the profusion of stimuli, however trivial or irrelevant, such as small food objects on a carpet, air or dust particles, details of the paper and print of a page that is being read, and so forth [53,65,77,78]. A possibly related abnormality concerns visual hypersensitivity. This involves excessive visual sensitivity to bright lights, the flicker of florescent lighting, and other background stimuli, and is frequently disturbing in ASD [63,79]. 

A prominent symptom is that ASD individuals commonly display apparent disinterest, or more precisely fail to display the usual enhanced visual interest, in most other people [58,61,80,81]. Kanner, for example, reported that ASD children commonly paid similar levels of attention to the humans in his office, as they did to the furniture, lights, and other inanimate objects. Nonetheless, recognition of faces and of other objects by ASD individuals is intact, so cannot account for this abnormality. This intact ability is evidenced by a comprehensive review of behavioral studies which found that ASD individuals process face recognition as TD individuals do, the results of earlier reviews, and subsequent empirical behavioral evidence [66,82,83,84]. It is also evidenced by neuroimaging findings in ASD that the fusiform gyrus (FG), a principal brain region that processes face recognition, has the capacity for TD magnitudes of activation, although it is normally hypoactive relative to TD [46,85,86,87,88,89,90].

A further common visual system abnormality is that ASD individuals relative to TD individuals preferentially recruit visual cortical areas for processing diverse cognitive tasks, such as the embedded figures task, arithmetic tasks, word learning tasks, and social understanding tasks, while achieving normative levels of behavioral performance (see for meta-analysis, [91]; see also [92]). 

Auditory abnormalities parallel those of the visual modality. Basic sensory function is generally normal in ASD according to audiometric tests [93,94,95], but high-level auditory abnormalities are frequent. Behaviorally, ASD children are frequently inattentive to significant sounds such as a parent’s voice or their own name, according to observational and experimental evidence [61,80,96,97,98]. Further, neural high-level processing of speech sounds in ASD groups does not display the enhancement of TD groups, whereas neural processing of matched control sounds is not significantly different between groups, as measured by event-related brain potentials or functional magnetic resonance imaging (MRI) [99,100,101,102,103]. ASD individuals commonly report poor filtering of irrelevant sounds, and their audition is generally indiscriminate and unselective. This leads to a profusion of stimulation for processing, and much distraction and disturbance from background sounds [61,62,63,65,104]. 

Nociceptive (pain) processing is commonly disturbed in ASD according to self-report, questionnaire, observational, and neuroimaging studies [25,81,104,105,106,107]. Self-report, observational, and neuroimaging evidence support commonly impaired sensitivity to conditions of heat and cold in ASD [30,53,58,65,108,109]. Numerous empirical studies have examined tactile hypersensitivity, hyposensitivity, and sensation seeking in ASD, predominantly using questionnaire and observational methods. Such studies have commonly reported tactile abnormalities, but findings are often inconsistent and inconclusive (see for review [110]). Difficulties in toileting are common in ASD [111], and poor viscerosensory awareness of bladder and bowel state contribute to such difficulties [61,65]. Impaired awareness of such bodily states as hunger and thirst are also experienced in ASD [104,112].

The proprioceptive system senses and represents position and movement of the body and its parts, and the vestibular system senses and represents absolute motion in space and position of the head and body [113]. Dysfunction of these systems is evidenced by reports of ASD individuals, questionnaire findings, and poor results on neurological tests [19,25,63,65,104,114,115,116]. Nevertheless, examination of proprioceptive performance on arm and fingertip tasks in ASD adolescents with confirmed movement impairments revealed intact low-level proprioceptive representations in this group [117], suggesting that impairments may be high-level ones. In sum, diverse sensory systems are commonly disturbed in ASD.

### 2.2. Motor Abnormalities 

Posture, gait, coordination, and gross and fine motor skills have been examined in neurological and experimental studies, and the findings are that there are substantial impairments in these domains in ASD relative to TD groups (see for review of motor skills [118]; see also [25,116,119,120,121]). In a study of bodily dimensions recruited in motor functions, ASD and TD subjects were required to estimate maximum arm reach, maximum hand grip aperture, and hand width, and these were compared to actual achieved values. The ASD group was significantly and markedly inaccurate compared to the TD group [122]. The action observation system, which is involved in the perception of others’ movements, was found not to be impaired in ASD [123]. Social and expressive gestures are often limited in variety, and may be performed infrequently, inappropriately, or poorly coordinated with other forms of expression [58]. In sum, there is substantive evidence of motor impairments in ASD, but there are numerous inconsistencies in the literature, and emerging confounds have often not been controlled [118].

### 2.3. Cognitive Abnormalities 

A major research topic has focused on theory of mind (TOM), which is the knowledge that individuals develop of their own and others’ mental states that facilitates understanding of behavior. A meta-analysis of such studies found a significant and large impairment of TOM in ASD adults versus TD controls [124]. A further facet of social cognition has been experimentally studied and is commonly impaired in ASD. Studies using silent film clips in which several geometric figures move and interact with one another [125] have found that ASD subjects’ reports on these film clips, relative to those of matched TD controls, are impoverished, omit a substantial proportion of social interpretations, describe fewer mentalistic or emotional items or use the corresponding terms inappropriately or irrelevantly, and observe fewer personality characteristics. Instead, ASD subjects’ reports mostly comprise physical-level cognition [126,127,128,129].

Abstract concept processing is frequently and significantly impaired in ASD. Grandin reported difficulties in mastering and understanding abstract concepts such as those of ‘over’, ‘liberty’, and ‘the future’ [62,130]. In studies of ASD individuals and matched TD controls, batteries of concept tests have been administered, and the findings were that aspects of concept processing were commonly impaired [27,131,132,133,134]. A likely related deficit is the inability of ASD individuals to generalize [27,132,134]. That is, a rule or solution that is learned in a particular situation, is rarely applied in different or novel situations that share common features with the learning situation, and where the rule could be applied [61,68,78,132,134].

Cognition at the concrete- or physical-level is generally preferred in ASD, and this is supported by the preferential processing of social situations at the physical-level reported above (see also [74]). Further, in a sorting task ASD children sorted more frequently on concrete features (color, size), whereas TD and ID comparison groups sorted more frequently on category membership (sports, games) [135]. Grandin’s cognition operates overwhelmingly in concrete visual images, and needs to transform abstract concepts into concrete visual images for processing and understanding [62,130]. Likewise, the increased recruitment of posterior visual regions in ASD relative to TD groups, to process diverse cognitive tasks [91,92], suggests enhanced use of concrete visual imagery to generate solutions. The concrete cognition preference may also develop into the circumscribed interests of ASD. These mostly comprise physical, mechanical, or rote knowledge, are frequently pursued with passion, accumulate vast amounts of factual information on the chosen topic, but rarely develop to higher-level conceptual knowledge, and may progress to the outstanding abilities of a substantial minority of ASD individuals [12,136,137,138,139,140]. Such concrete preferences may commence early in life, as ASD toddlers were found overall to visually fixate for longer on moving geometric images than on moving social images, and for longer than did matched toddlers with TD, developmental delay (DD), other developmental conditions, and unaffected siblings of ASD probands [141,142]. 

Specific memory functions may be impaired in ASD. Tasks with requirements for memory for faces or for words were found by systematic reviews and meta-analysis to be impaired in ASD compared to TD groups [84,124]. Everyday memory refers to the types of memory demands that occur in real life, such as remembering where an object was previously placed, or to post a letter on encountering a post box. Such memory was assessed with a number of subtasks of the Rivermead Behavioral Memory Test, and ASD individuals were found to be significantly impaired on a number of them, particularly those that tapped prospective memory (i.e., remembering to remember) [143]. In sum, multiple forms of cognition are commonly impaired in ASD. 

### 2.4. Emotion Function 

ASD individuals experience diverse emotions, suggesting the capacity for subjective emotions is intact, but emotions can contribute to symptomatology. The subjective emotions of pleasure, joy, and enjoyment are recorded in self- and clinical reports of ASD individuals [61,80,138]. An empirical study examined by self-report questionnaire ASD adolescents and matched TD controls, and found that the groups did not differ on subjective pleasure ratings elicited by physical or other sources (e.g., eating, scents, achievement of a goal), but both groups reported social interactions to be mildly unpleasant, the ASD group significantly more so than the TD controls [144].

Love is experienced by ASD individuals, according to self and observer reports [61,138]. For instance, an ASD child with an intense circumscribed interest in watches stated that he would give away his favorite watch for a girlfriend [138]. A study that used parent completed questionnaires found that ASD adolescents and young adults sought romantic relationships, but often pursued them with inappropriate people (e.g., celebrities), and often with inappropriate forms of courtship [145]. In regards to friendship, a meta-analysis was performed on studies of friendship in ASD, that used self-, peer-, and parent reports. It was found that ASD boys do wish for friendships and successfully form friendships, but these are fewer in quantity and lower in quality than those of TD controls [146]. 

Loneliness is an emotion that arises from deprivation of close loving relationships, or of a social network [147]. An investigation of loneliness in ASD children and adolescents used observation, social understanding testing, a self-report questionnaire, and a structured interview. In response to specific questions on loneliness, a majority of ASD subjects demonstrated good understanding of this emotion, and the ASD group reported significantly higher levels of subjective loneliness than the TD group [147]. An interview study examining a wider range of emotions found substantial magnitudes of loneliness to be frequent in ASD [22], and in a self-report questionnaire study findings of elevated loneliness in ASD were again reported [148]. Thus, ASD individuals experience subjective love and liking, and consequent loneliness when deprived of them.

Perceived stress levels, as assessed by questionnaire, have been found to be significantly and greatly elevated in ASD compared to TD controls [149,150]. A further study measured cortisol levels in children’s hair, and found significantly higher levels in the ASD compared to TD group, which is evidence of chronic stress [151]. Anger, rage, and frustration are reported in ASD [61,80]. Substantial magnitudes of irritability and temper tantrums were reported by parents to be frequent in ASD adolescents (prevalence of 55% and 25%, respectively) [22], and physical aggression prevalence was reported at 30–56% in ASD samples [13,14,22].

Fear was reported by Jolliffe to dominate her life [61] (p. 16), and was elicited by diverse social and non-social stimuli. Grandin also observed she was much troubled by severe fear and anxiety [62]. The prevalence of any clinical anxiety disorder has been reported at 42–84% in ASD samples [152,153,154,155]. Findings are somewhat inconsistent, in part for methodological issues [156].

Depression, despair, distress, and misery are further dysphoric emotions experienced in ASD [22,61]. The prevalence of clinical depression in ASD samples has been found to range from 30–50% [22,152,153,154], and in an adolescent ASD sample 40% suffered ‘chronic unhappiness’ according to parent report [22]. Collectively, many subjective emotions are intact and functional in ASD, but some are abnormally intense and impairing.

### 2.5. Repetitive Behaviors 

A diverse set of behavioral abnormalities is collected together into the domain of restricted repetitive patterns of behavior, interests, or activities [58]. Examples are pacing and stereotyped walking; rocking of the body; inflexible rituals that are integral to washing, dressing, and other regular activities; insistence on sameness and intolerance of even trivial changes to environment, routines, or rituals; circumscribed interests as described earlier; and SIBs such as hand biting or head banging [58,107,157,158]. Models involving two or three subtypes of repetitive behaviors have been generated [158,159,160], and in support of these developmental patterns, and associations with ASD and patient variables, differ across subtypes [158,159]; such models have value but also limitations. Repetitive behaviors appear early in life, many of them are persistent, and obstruction of them can elicit severe reactions, including ‘meltdowns’. They are also one of the most difficult and disrupting aspects of the disorder for caregivers [138,157,160,161,162]. 

### 2.6. Daily Living Skills Impairments 

ASD individuals are frequently and disproportionately impaired in self-care and other activities for self-sufficient living [16,163,164]. Feeding, for instance, is often disturbed; choice of what to eat is particularly narrow and inflexible [58,165,166], and interoception of hunger and thirst is often impaired [104,112]. Self-care difficulties in toileting, washing, and dressing are reported. Poor awareness of bowel and bladder state, and impaired representation of body boundaries and of body size relative to the size of a lavatory, have been reported [61,65]. In addition, Jolliffe reported an inability to match each shoe to the corresponding foot, and seemed to have difficulty representing and processing such body parts [61]. Williams reported dyspraxia of dressing, finding it difficult to put on clothing appropriately [65]. A further self-care deficit is poor understanding of the dangerousness of objects and conditions. ASD individuals may walk on high ledges or other hazardous places, in front of cars, or climb high into trees, with little apparent awareness of the dangerousness of their situation [40,53,107,167]. ASD individuals frequently have difficulties in planning, decision-making, and implementation of such daily living activities such as selecting clothing that is appropriate for weather or social conditions, making telephone calls, using a restaurant, handling money, or using public transport [22,108]. A meta-analysis of experimental studies that investigated planning skills found significantly but moderately reduced performance in ASD compared to TD controls [168]. Together, living skills are commonly compromised in ASD, and likely entail multiple impairments.

Medical disorders are common in ASD, with one study finding that only 29% of ASD individuals had excellent health according to parent reports [169]. Gastrointestinal (GI) disorders, sleep difficulties, and epilepsy occur at elevated rates in ASD [58,169,170,171,172,173]. Enlarged head circumference (macrocephaly) was found in 16% of ASD individuals but 3% of comparison individuals, and enlarged total brain volume measured by structural MRI was found in 9% of ASD individuals in meta-analyses of relevant studies [174].

### 2.7. Social Impairments

Social impairments are prominent features of ASD, and some of these have been summarized in the sections above. ASD individuals commonly have difficulty making sense of social situations, do not understand social cues, nor social conventions, and manifest impaired social cognition [58,61,62,65,127,128,129]. They may treat fellow humans similarly to other objects, are apparently disinterested in them or in forming relationships with them, often look through acquaintances despite intact face recognition, and do not prefer their mother’s voice over control noises [58,61,80,84,96]. Such deficits likely compromise the development of joint attention and other social processes, that foster diverse forms of learning. Social behaviors may be inappropriate, such as failing to respect personal space, making candid comments, and contravening social conventions or ethics [58,61,62,63,80,175]. 

Many language skills in ASD, such as grammar and vocabulary, range along a continuum from absence of them to largely typical levels of competence [9,10]. Pragmatic aspects of language are almost universally impaired in ASD. Speech delivery is often monotonous or machine-like, volume is often poorly controlled, stress patterns and other components of prosody are poorly expressed and poorly interpreted, rhythm more generally may be impaired, expressive gestures and facial expressions are limited, non-literal language such as metaphor, sarcasm, and jokes are not comprehended, and conversation tends to dwell endlessly on circumscribed interests irrespective of conversational partners’ responses [9,10,58,176,177,178,179,180]. Moreover, social impairments start to emerge early in life [15,18,26]. Collectively, social interactions in ASD are commonly impaired in multiple ways, and these impairments likely foster stress and anxiety, withdrawal, further dysphoric emotions, and adverse health effects (see Section 5.3; [58,61,62,149,150,151]). 

### 2.8. Summary and Proposed Pathogenic Mechanism 

ASD symptomatology comprises sensory, motor, cognitive, emotional, repetitive behavior, difficulties in daily living, social, and language categories of symptoms, so extends well beyond just social symptoms. Particular symptoms may occur frequently but not universally across ASD individuals, and conversely the pattern of symptoms across ASD individuals is enormously heterogeneous [8,181]. Further, many studies have reported contradictory findings, in part for methodological reasons. In addition, some symptoms are under-studied, as with abnormalities of interoception of hunger, thirst, body temperature, and other bodily variables, as well as impairments of body representation. Taken together, much has been learned but more remains to be learned about the symptoms and features of ASD.

There is, however, likely a common pathogenic mechanism of the disorder [36,37,38]. In the following sections, disrupted neurocircuitry is first addressed. Four social brain regions, the amygdala, OFC, TPC, and insula, are disrupted in ASD and supporting evidence is summarized; these constitute the proposed common pathogenic mechanism of ASD. Symptomatology is then addressed: widespread ASD symptoms can be explained as direct effects of disrupted social brain regions. Sequelae of these disruptions and dysfunctions can explain many further ASD symptoms, and relevant evidence is summarized. Together, the four social brain regions, their disruptions, and sequelae provide an extensive account of ASD symptomatology. This model is summarized in Figure 1 and Figure 2; the disrupted neurocircuitry is summarized in Figure 1, and the resulting symptomatology is summarized in Figure 2.

## 3. Neurocircuitry: The Four Social Brain Regions are Commonly Disordered in ASD

### 3.1. The Amygdala is Disordered in ASD 

See Figure 1, Box 1. 

The amygdala is a structure of some 12 million neurons in humans that is situated in the temporal lobe in an anterior, medial, and ventral location. The nuclei particularly involved in ASD are the lateral, basal, and accessory basal nuclei, which account for 33%, 27%, and 10% of amygdala cells, respectively [182,183]. The amygdala is generally disordered in ASD, as evidenced next. 

Functional neuroimaging investigations have assessed brain function during face processing and other social tasks, and the findings were that the amygdala was hypoactive in ASD individuals relative to TD controls (see for meta-analysis, [46]). A further neuroimaging paradigm examined resting state functional connectivity. It found significantly reduced resting functional connectivity that predominantly involved the amygdala, insula, and OFC in ASD adults relative to TD controls [184]. Amygdala disruption is likely underestimated by fMRI studies, however, due to multiple technical issues, such as magnetic-susceptibility-induced signal loss, and individual differences variables [185].

In a single cell recording study of two rare neurosurgical patients with ASD and eight non-ASD control patients, testing was carried out on 37 and 54 amygdala neurons, respectively. It was found that in the ASD patients, basic electrophysiological measures of neural function were essentially normal. An exception, however, was that functional abnormalities were found in a sub-population of amygdala cells; selectivity for the mouth region was significantly increased, and for the eye region was significantly reduced in the ASD patients compared to control patients [186]. In another electrophysiological experiment that involved a severely autistic boy, electrodes were placed in several amygdala nuclei bilaterally and in interconnected nuclei (e.g., bed nucleus of the stria terminalis), and a program of deep brain stimulation (DBS) was applied. The results were substantial and persistent amelioration of multiple ASD symptoms, and were driven by stimulation specifically of basolateral amygdaloid nuclei [187]. 

Structural neuroimaging studies report that amygdala volume of young ASD children is significantly enlarged relative to that of young TD children. In adolescence and adulthood, however, amygdala volume of ASD individuals is similar in magnitude or somewhat smaller than that of healthy controls [45,47,48,188,189,190,191]. In a subregion level study, amygdala enlargement in ASD children was localized to the laterobasal subregion, comprising lateral, basal, and paralaminar amygdaloid nuclei [188]. Moreover, amygdala volume abnormalities are related to the level of ASD symptoms [188,192,193]. In a longitudinal study of 3–6 year old children with ASD, larger right amygdaloid volume, measured by structural MRI, was predictive of greater current and subsequent deficits in social interaction and communication skills [192]. Conversely, in adult males with ASD, smaller amygdaloid volume was associated with greater social impairment, as indexed by discrimination of emotional from neutral facial expressions, and eye fixation time [193]. Thus, the behavioral impairments correspond with the trajectory of amygdala volume abnormalities further supporting an amygdala contribution to ASD.

A cellular study has applied modern stereological, quantitative methods to post-mortem brains to assess any changes in the ASD amygdala. The major findings were that the ASD amygdala overall comprised a significantly lower number of neurons relative to controls, but there were no differences in amygdala neuron size. As to amygdaloid nuclei, the lateral nucleus comprised a significantly lower number of neurons, but for other amygdaloid nuclei neuron numbers were non-significantly lower [183]. In a further quantitative stereological study, the lateral amygdaloid nucleus was again the most abnormal nucleus, with neuron numbers being reduced by 17% in ASD compared to control subjects, but this difference was not statistically significant. Neuronal numerical density in this nucleus, however, was significantly reduced [49]. In summary, there is substantial and convergent evidence that the amygdala is commonly disordered in ASD. 

### 3.2. OFC is Disordered in ASD 

See Figure 1, Box 2.

OFC is located in the anterior ventral part of frontal cortex, and comprises Brodmann’s areas (BAs) 10, 11, 12 (also called 47/12) [194], 13 (part of which is insula), and 14 [195]. (A diagram summarizing all Brodmann’s areas is presented in [196]). OFC is commonly disrupted in ASD, as evidenced next.

OFC is difficult to neuroimage accurately due to signal dropout, geometric distortion, and susceptibility artifacts [197,198,199], hence OFC abnormalities may be underreported. Notwithstanding, functional neuroimaging studies have overall reported hypoactivation of OFC (BA 47/12) in ASD, according to the meta-analysis of Patriquin et al. [46]. In a further neuroimaging study not included in the above meta-analysis, film clips of biological motion were presented to three groups: ASD children, their unaffected siblings, and unrelated TD children. The findings were that OFC (BAs 10, 11) was hypoactive in ASD relative to the comparison groups [200]. Further, in the comparison group of unaffected siblings of ASD children, OFC (BA 11) and superior temporal sulcus (STS; BAs 22, 39) activations were increased relative to the other groups. This suggests OFC can mediate compensatory processes in ASD [200]. This is further suggested by findings that OFC volume was enlarged in two patients with amygdala lesions caused by Urbach–Wiethe disease [201]. In a different paradigm that used functional MRI and seeded-regions based connectivity analysis, the findings were of significantly reduced resting functional connectivity among OFC (BA 10), amygdala, and insula in ASD relative to the TD group [184].

In regards to structural abnormalities, a meta-analysis of structural neuroimaging studies of ASD reported a small but significant increase in grey matter volume in BA 10 and adjacent BA 46 ([48]; but see [45,47] for null results). In a later structural neuroimaging study that controlled for such confounds as language impairments, medication use, and comorbid disorders, OFC (BA 11) volume was significantly reduced in ASD relative to the TD group [190]. In sum, there is evidence of OFC disruption in ASD, but this may be underestimated due to technical issues in MRI neuroimaging and OFC compensatory plasticity in some ASD individuals.

### 3.3. TPC is Disordered in ASD 

See Figure 1, Box 3.

TPC broadly comprises superior temporal regions and adjacent inferior parietal regions, and is generally taken to encompass BAs 41, 42, 22, 43, 40 (supramarginal gyrus (SMG)), and 39 (angular gyrus (AG)). TPC subregions are commonly disrupted in ASD, as evidenced next.

In a meta-analysis of functional neuroimaging studies, the findings included that the superior temporal gyrus (STG; BA 22) and inferior parietal cortex (BA 40) were hypoactive in ASD groups relative to TD comparison groups [46]. In a further meta-analysis that concerned language processing tasks, hypoactivation of middle temporal gyrus (MTG; BA 21) across diverse language tasks, as well as abnormalities in STG (BA 22), were found in ASD groups relative to TD controls [202]. In addition, a magnetoencephalographic (MEG) study found the perisylvian cortex, which includes TPC, to frequently display epileptiform activity in ASD children [203]. A further observation noted earlier is that in a group of unaffected siblings of ASD children, STS (BAs 22, 39) manifested increased activations relative to the two comparison groups, suggesting that STS can mediate compensatory processes in ASD [200]. 

In regards to structural neuroimaging studies, meta-analyses have reported findings of abnormalities in parietal operculum in the inferior parietal cortex, several further parietal subregions, as well as the MTG [45,48]. A subsequent surface-based morphometry study that focused on social brain regions found reduced cortical surface area of the superior temporal cortex in ASD [46]. In addition, the Sato et al. [190] structural study reported significantly reduced grey matter volume of BAs 21 and 22 in ASD compared to TD controls. Taken together, there is substantive evidence for TPC abnormalities in ASD. 

### 3.4. Insula is Disordered in ASD 

See Figure 1, Box 4.

The insula is the cortex located in the depths of the Sylvian fissure, and is covered by the orbitofrontal, frontoparietal, and temporal opercula (flaps of tissue that hide the fissure). It is bounded by the sulcus circularis, and is comprised of anterior and posterior lobes. It is further divided into subregions, which vary across parcellation schemes. An influential scheme recognizes agranular, dysgranular, and granular subregions, which are located in anteroventral, mid, and posterodorsal insula, respectively [204]. The insula’s involvement in ASD has been insufficiently examined [45]; nonetheless, it is likely disrupted in ASD, as evidenced next. 

In a meta-analysis of functional neuroimaging studies, it was found that the insula (BA 13) was significantly hypoactive in ASD relative to TD comparison groups [46]. In a further paradigm that used neuroimaging and seeded-regions based connectivity analysis, the findings were of significantly reduced resting functional connectivity among the insula, OFC, and amygdala in ASD relative to the TD group [184]. 

Regarding structural studies, a meta-analysis of structural neuroimaging findings reported significant volumetric abnormalities in the insula and adjacent parietal operculum in ASD relative to TD controls [45]. In addition, a surface-based morphometry study of social brain regions found reduced cortical surface area of the insula in ASD [46]. Thus, despite limited research interest, there is convergent evidence of insula disruption in ASD.

### 3.5. Summary 

The disruption of four social brain regions, the amygdala, OFC, TPC, and insula, in ASD has now been substantively evidenced. Some of these regions, however, have been much researched (e.g., amygdala), but others somewhat neglected (e.g., insula), so the weight of evidence varies. The functions of the four social brain regions, and what their disruptions directly contribute to ASD symptomatology, are set out in the next section. Numerous sequelae of such symptoms and disruptions are set out in the subsequent section. Together, a high proportion of ASD symptoms and features can be so explained. 

## 4. Symptomatology: The Four Social Brain Regions’ Multiple Functions, and Their Direct Contributions to ASD Symptoms and Features

### 4.1. Amygdala Disruption Likely Underlies Specific ASD Symptoms 

See Figure 2, Box 2.

The amygdala mediates numerous functions, which are not yet comprehensively understood. Several amygdala functions have been heavily researched, particularly the amygdala’s contributions to fear, and to emotional memory enhancement, and are extensively reviewed elsewhere (e.g., [205,206,207,208,209]). The former function, however, likely requires further specification, because multiple studies have reported that individuals with amygdala lesions can continue to experience fear [210,211,212,213,214]. This suggests that the amygdala represents only components of fear, which have yet to be clarified (cf. [215]) and whose contributions to ASD symptomatology are unclear. The latter function likely contributes to the ASD symptom of impaired memory ([143]; see Section 5.4.4). 

Further amygdala functions likely remain to be discovered (e.g., [216]), but further principal functions that have long been proposed but less investigated are the representation of diverse forms of intangible knowledge [217,218]. Disruption of such representations is proposed to contribute to multiple prominent ASD symptoms. Intangible features of stimuli have been insufficiently researched, so there is no accepted definition of them. A working definition is that they are “non-physical or non-concrete properties of stimuli, that commonly and substantively modulate behaviors and cognition” [219]. Intangible knowledge representations with known amygdala involvement include: impact, importance, exclusiveness, noxiousness, valence, economic value, same group membership, social status, social popularity, trustworthiness, features of morality, ambiguity, dangerousness, relevance, and unpredictability. These hypotheses are predominantly evidenced by findings from human neuropsychological and neuroimaging work (see for extensive review, [219]; see also, [220,221]).

A characteristic and widespread visuosocial feature of ASD individuals is the atypical and disorganized visual scanpaths they display [76]. In TD individuals and monkeys, recordings of visual scanpaths reveal that salience, valence, arousal, and likely related features, are major factors in organizing the visual scanpaths they execute in viewing social scenes [67,70,71,72,73,222]. In ASD individuals, visual scanpaths are commonly atypical, and there is greater guidance from physical features, their recognition of objects and faces being essentially intact [66,74,75,76,82,84]. A detailed behavioral study examined the processing stages during which the ASD atypicalities arise during the free viewing of complex naturalistic scene images. The findings were that the early stages of visual processing, involving processing at the basic visual and object levels, were not significantly different, but later stages, particularly involving the processing of meaning, differed significantly between ASD and TD groups [75]. Thus, in the organizing of ASD compared to TD scanpaths, there is lesser involvement of some meaningful features, but greater involvement of physical features. At the neural level, the amygdala relays robustly to BA 45B, which projects to the frontal and supplementary eyefields (FEF, SEF) and participates in the high-level guidance of visual gaze [223]. In addition, a monkey study of eye gaze used single cell recording and eye tracking to measure processing of natural images. It found that the amygdala’s intense activations to pixels in social images were strong predictors of eye fixation hotspots during free viewing [222]. Taken together, intangible knowledge participates in the organizing of eye gaze scanpaths in TD individuals. Moreover, the amygdala represents forms of intangible knowledge, participates in the structural neural network that guides eye gaze, and participates in functional eye gaze processing. Thus, amygdala dysfunction is proposed to be involved in the disorganized visual scanpaths of ASD. Some preliminary evidence is provided by findings that rewarded face stimuli influenced visual attention of TD preschoolers significantly more strongly than that of ASD preschoolers [224]. 

A further visuosocial atypicality is that ASD individuals fail to express heightened interest in fellow humans, including close family members [58,61,80]. At the neural level, this symptom is hypothesized to be mediated by failure to enhance activation of a functionally intact visual cortex, including its FG subregion, by the impaired amygdala. Structurally, the visual cortex receives heavy projections from amygdala, which have an excitatory effect on it, and they likely engage in interactive processing of facial expressions and other significant stimuli [182,225,226,227,228]. In ASD, FG is commonly hypoactive relative to TD groups during the performance of social tasks, but it can be activated to typical levels by suitable stimuli, so is functionally intact [46,85,86,88,89,90]. The amygdala is impaired, and it has reduced connectivity with FG [229,230], and these factors likely result in failure to enhance activation of intact visual cortex to significant stimuli. Consistent with this, monkeys with experimental amygdala lesions and a normal visual cortex failed to enhance activation of visual cortex to facial expressions, whereas control animals did so [231]. Amygdala dysfunction and visual cortex hypoactivity are hypothesized to manifest as a lack of heightened interest in important social stimuli such as fellow humans and close family members.

Corresponding atypicalities occur in the auditory modality. ASD children frequently fail to express heightened interest in significant sounds such as their own name, a parent’s voice, and language, whereas TD children commonly do so [96,97,98,100,102]. At the neural level, amygdala dysfunction may drive these ASD atypicalities also. The auditory cortex is heavily interconnected with the amygdala, and in ASD it is hypoactive to significant social stimuli, but is likely functionally intact as activation is normal to control stimuli, and audiometric tests are commonly normal [93,94,95,99,100,101,102,182,232]. Auditory cortex hypoactivation to significant stimuli driven by amygdala disruption is suggested to manifest as the atypical lack of heightened interest in such stimuli in ASD.

ASD individuals express an inability to understand social conditions [61,62,65]. Impaired intangible knowledge is hypothesized to contribute to this deficit, as may impaired conceptual cognition (see Section 5.1). The amygdala also participates in the processing of joint attention [233], and its dysfunction likely impairs this processing.

Sensory hypersensitivity and hyposensitivity are common ASD features which frequently co-occur, and likely involve shared mechanisms [58,61,68,95,110,234]. Klin et al. [68] contend that importance or salience enable complex conditions and environments to be rapidly discriminated into those components that merit processing (e.g., important ones), and those that do not merit processing (the insignificant, trivial, background ones), and that without this, the usual profusion of stimulation would be processed indiscriminately, so would be overwhelming. In ASD, there is indiscriminate and unorganized processing of environments in the visual and auditory domains, which results in stimuli that merit increased processing being relatively neglected (hyposensitivity), while trivial and background stimuli are processed to an abnormal extent [53,61,62,63,65,77,78,104,176]. In addition, the profusion of stimulation that elicits processing may account for the perception of it as excessive and overwhelming (hypersensitivity; [61,62,65]; cf. [68]). Building on Klin et al. [68], it is hypothesized that these effects may be ascribed to impaired intangible cognition, arising from amygdala dysfunction. 

Amygdala dysfunction likely contributes to ASD impairments in self-sufficient living. The amygdala participates in the processing of planning [216,235], and planning is engaged in self-sufficient living [22,236,237,238]. Hence, amygdala dysfunction in ASD likely contributes to disrupting such self-sufficient living tasks as buying goods, using public transport, managing finances, and so forth. The amygdala is engaged in the networks that represent harmfulness and trustworthiness [219,239], and such knowledge is disrupted in ASD so likely increases perilousness and vulnerability in daily living [40,53,58,107,167]. In summary, amygdala disruption likely gives rise to multiple prominent social as well as non-social symptoms of ASD.

### 4.2. OFC Disruption Likely Underlies Specific ASD Symptoms 

See Figure 2, Box 3. 

OFC dysfunction impairs intangible knowledge representations of rightness, wrongness, and appropriateness. For instance, patients with OFC lesions are disrespectful of authority, uninhibited, frequently use foul language, steal, lie, and endorse forms of immoral actions that are rejected by healthy comparison groups [237,238,240,241,242,243]. OFC dysfunction likely participates in the limited ability of ASD individuals to comprehend that some behaviors are socially wrong or inappropriate, such as aggressive forms of courtship, and tactless or rude comments about others [63,145,175]. OFC participates in processing joint attention [233], and its dysfunction likely impairs such processing. OFC dysfunction likely impairs the high-level multimodal food palatability representations that OFC normally mediates [198,244,245,246,247]. This likely disables flexible food choice processing, with consequent dependence on inflexible habit-based processes [248], hence the widespread ASD symptom of narrow and inflexible food selectivity [58,166,167]. OFC lesions are found by neuropsychological studies to be associated with severely impaired planning, and processing and representation of decisions [199,236,237,238,240,241,248,249]. These impairments likely participate in ASD individuals’ difficulties in planning and decision-making, which also contribute to disrupting daily living activities, including such routine tasks as choosing appropriate clothing, making telephone calls, using public transport, managing finances, and so forth [14,22,108]. OFC participates in the regulation of anger, aggression, and sleep, and lesions and dysfunction of it are associated with poor regulation of these functions [241,250,251,252,253,254,255,256]. These dysregulations likely contribute to the ASD features of elevated intensity and frequency of anger and aggression, and elevated rates of sleep difficulties [13,14,22,170,173]. In sum, it is hypothesized that OFC disruption likely contributes to multiple specific ASD symptoms and features. 

### 4.3. TPC Disruption Likely Underlies Specific ASD Symptoms 

See Figure 2, Box 5. 

TPC subregions that integrate multisensory information (temporoparietal junction, parieto-insular vestibular cortex, inferior parietal lobe, and SMG) variously participate in networks that process the maintenance of body posture [257], and in the planning and performance of skilled movement [258,259,260]. Dysfunction of such TPC subregions likely participates in ASD individuals’ unusual posture, odd gait, clumsiness, and other motor abnormalities. Consistent with this, these symptoms have been ascribed to disrupted multisensory integration, specifically of visual, proprioceptive, and vestibular inputs, or to disruption of the proprioceptive sensory system, or of other motor-related circuits [116,119,120,261]. TPC is also hypothesized to be an integrative hub region that mediates abstract representations of social and other contexts [262]; see for reviews, [263,264]. TPC disruption may thus contribute to ASD difficulties in understanding different social contexts. TPC subregions (BAs 22, 39) participate in a network for moral cognition processing, and this is evidenced by meta-analyses of neuroimaging studies [265,266]. Thus, TPC dysfunction likely contributes to the impaired understanding of rightness, wrongness, and appropriateness reported in ASD [63,145,175]. 

The TPC is a multimodal integrative region that importantly participates in the networks that mediate body representations, and this is evidenced by the findings of neuroimaging, electrical stimulation, and neuropsychological studies in human [267,268,269,270,271,272,273,274]. Further, networks for body representations overlap those for motor functions, and this is supported by electrophysiological findings in humans [275]. Impaired body representations are associated with dyspraxia of dressing, related daily living impairments such as toileting difficulties, and likely inaccurate motor planning [65,122,270,276], so TPC dysfunctions may participate in the corresponding ASD symptoms. Notwithstanding, impaired body representation in ASD is supported only by anecdotal reports of ASD individuals (see symptoms summary in Section 2.6), some empirical evidence [122], but little formal study.

The TPC participates in the representation and processing of rhythm, musical expressiveness, prosody, and expressive behaviors [276,277,278,279,280,281,282,283,284], as well as plays a major role in language processing [285,286,287,288,289]. TPC dysfunctions are thus hypothesized to contribute to the ASD symptoms of poor rhythm, monotonous and unexpressive speech, poor prosody, impoverished and poorly coordinated expressive behaviors, as well as further language deficits [9,58,176,179,202]. In sum, it is hypothesized that TPC disruption likely participates in multiple specific ASD symptoms and features. 

### 4.4. Insula Disruption Likely Underlies Specific ASD Symptoms 

See Figure 2, Box 6. 

The insula participates in numerous sensory processing networks, and this is evidenced by findings of neuroanatomical, electrical stimulation, and neuroimaging studies in monkeys and humans [204,290,291,292]. Further, patients with lesions of the insula have been found to suffer from impaired sensory processing in somatosensory, thermosensory, nociceptive, gustatory, viscerosensory, and vestibular modalities [293,294]. Impaired processing of diverse sensory modalities is reported in ASD, and these commonly include somatosensory, thermosensory, nociceptive, viscerosensory, and vestibular impairments (see Section 2.1), so these may originate with insula disruption. These impairments may participate in such ASD abnormalities as unresponsiveness to injuries, poor awareness of hunger, thirst, body temperature, and bowel and bladder state [61,63,65,104,106,107,108,112]. The insula participates in behavioral responses to homeostatic challenges such as hunger, thirst, cold, or hot conditions [295,296,297]. Insula dysfunction may contribute to impaired homeostasis-related and body maintenance behaviors that are reported in ASD, such as failure to prepare for cold conditions, wearing of unsuitable clothing, impaired eating and drinking habits, and so forth [58,65,108]. The insula importantly participates in forms of body representation [268,271,272,273], so its dysfunction may contribute to body representation and related impairments summarized in the TPC section above. The insula participates in aspects of language and speech processing, such as recognition of sounds, and production, comprehension, and pronunciation of speech [204,288,291,293] so its dysfunction may participate in the speech and language abnormalities that are widespread in ASD [9,10]. The insula is a major component of the network that represents and processes expressive behaviors [281], so its dysfunction may contribute to the ASD symptom of impoverished and uncoordinated expressive behaviors [58]. In sum, it is proposed that insula dysfunction likely participates in multiple specific ASD symptoms and features.

## 5. Sequelae 

See Figure 2, Boxes 7–9.

The ASD neural disruptions and symptoms so far described can secondarily produce further symptoms and features. Specifically, disrupted brain regions can drive disruption of strongly interconnected brain regions to produce further symptoms, and particular symptoms can drive further symptoms or adaptive responses [50,51,52,53]. Such sequelae constitute a substantial proportion of ASD symptomatology, and are summarized in this section. 

### 5.1. Cognitive Abnormalities 

See Figure 2, Box 7.

Disrupted brain systems are associated with enhanced development and functioning of intact brain systems, and this effect has been found across multiple brain regions that were damaged by diverse brain diseases and lesions [298,299,300,301,302,303]. Correspondingly, ASD individuals who are hypothesized to have weakness in intangible knowledge will predominantly process concrete-level knowledge thereby becoming specialized and unusually skilled in this domain. There is abundant evidence of preference for and strength in concrete-level cognition in ASD (see symptoms summary in Section 2.3). Thus in this disorder, weakness in intangible cognition is proposed to bring about the prominent but unexplained emphasis on concrete-level cognition.

Concrete-level cognition, however, produces adverse consequences. It has been suggested to impair the development of abilities in conceptual cognition and in the ability to generalize across contexts, which are ASD features [62,78,134]. Consistent with this, in a study using the Verbal Concept Formation Task, it was found that grouping of objects on the basis of concrete characteristics (e.g., oranges and bananas both taste sweet), rather than on the basis of meaningful, abstract characteristics (e.g., oranges and bananas are both fruits), was associated with impaired conceptual processing in patients with degenerative disease affecting frontal cortex [304]. Further, concepts (e.g., dog, fruit, car) are crucial in cognition [305]; they organize much enduring knowledge, and they facilitate efficient performance of diverse forms of cognition. Without concepts, much information would require to be repeatedly re-discovered [305]. Impaired concepts likely contribute to ASD individuals’ struggle to understand the stimuli and events that surround them [61,62,65].

The multiple cognitive shortcomings of ASD likely elicit adaptive responses. These are suggested to manifest in the ASD features of rote learning, use of formulae, rigid repetitive routines and rituals, and insistence on sameness [58,61,157]. As Jolliffe et al. [61] (p. 16) report: “Reality to an autistic person is a confusing, interacting mass of events, people, places, sounds and sights. There seem to be no clear boundaries, order or meaning to anything…. Set routines, times, particular routes and rituals all help to get order into an unbearably chaotic life….” 

### 5.2. Daily Living Skills Deficits 

See Figure 2, Box 8. 

Substantial deficits in daily living skills are a heavy concern for ASD individuals and their carers, but are not explained by social difficulties, motor difficulties, nor IQ, and their mechanisms are currently unclear [14,16,21,22,23,58,163,164,306]. Instead, they may be sequelae of dysfunctions of the four social brain regions, and of sensory disruptions in ASD. The insula participates in visceroception and in diverse homeostatic processes, so its dysfunction may contribute to self-care impairments in eating, drinking, thermoregulatory behaviors and related clothing choice, unawareness of the state of the bladder and bowel leading to toileting dysfunctions, as well as unresponsiveness to injuries [53,58,61,65,81,104,108,112]. Insula and TPC disruptions commonly produce impaired body representations, which are associated with difficulties in dressing, toileting, and likely other self-care activities [61,65,270,276]. The OFC mediates high-level food palatability representations [244,245,246,247], and its disruption likely contributes to limited and repetitive food choices in ASD [165,166]. OFC disruption impairs representations of wrongness and inappropriateness, as well as regulation of anger and aggression [198,237,238,251,252]. Such impairments can harm social relations, employment, and other aspects of self-sufficient living [63,145,175,236,237,238]. OFC and amygdala disruption impair planning and decision-making, which are impaired in ASD and seriously disturb widespread aspects of daily living [22,168,236,237,238]. In sum, sequelae of all four dysfunctional social brain regions are likely involved in impaired self-sufficient living, and this multiplicity of contributions may account for the frequency and magnitude of such impairments. 

### 5.3. Emotional Abnormalities 

See Figure 2, Box 9. 

The diverse dysphoric emotions commonly experienced in ASD are suggested to be substantially driven by ASD symptomatology, but ameliorated by stereotypies and SIBs. Elevated levels of stress are reported in ASD [149,150,151], and these are substantively driven by social difficulties according to self-reports, and were strongly correlated with autistic traits in a questionnaire study [61,62,65,150]. At the same time, perceived stress may impair social functioning [149]. Frustration and anger are commonly driven by such ASD features as difficulties in speech or other means of expression, or disturbance of rituals, repetitive activities, circumscribed interests, desired sameness, or SIBs according to self or observer reports [13,61,157]. OFC dysfunction in ASD likely exacerbates these emotional abnormalities. 

Fear, anxiety, and withdrawal are elicited by the ASD features of impaired understanding of social situations, impaired social and communicative functioning in social networks, as well as co-occurring social factors such as increased bullying, and numerous non-social stimuli [61,62,65,307,308,309,310,311,312,313]. Loneliness has been found to be elicited by the ASD features of poor quality of the best friendship and impoverished social networks [148,309]. Reported contributors to depression in ASD are: stressful life events such as parental disputes, divorce, or death, their effects likely amplified by ASD individuals’ dependence on support; ASD individuals’ subjective perceptions of being different, handicapped, or possessing shortcomings; cherishing hopes for social relationships and social networks that peers enjoy but finding them unattainable or unsatisfactory; suffering elevated levels of bullying and teasing; experiencing elevated levels of irritability, fear/anxiety, and loneliness which are risk factors for depression; and so forth [148,309,312,313,314,315,316,317,318,319,320,321].

Relief of dysphoric states and emotions is sought through the performance of repetitive stereotyped movements and SIBs in a wide range of animal species, and these types of abnormal behaviors are often linked [322,323,324,325,326]. For instance in monkey studies, performance of a bout of SIB was found to be preceded by stresses and stress responses, then succeeded by lowering of heart rate and stress hormone responses [323,324]. In humans, self-report, diary, interview, and behavioral studies of individuals who perform SIBs have found that levels of dysphoric emotions are elevated, and that relief of dysphoric emotions is the principal factor that drives SIBs, whereas self-punishment and social factors are lesser contributors [327,328,329,330,331,332,333,334]. 

In ASD, heightened dysphoric emotions and other features contribute to stereotyped behaviors and SIBs. The dysphoric procedure of blood draws carried out on children with ASD and ID was followed immediately by increased rates of SIBs in some participants, suggesting a cause-effect relationship [107]. Increased levels of maternal criticism were found to predict increased behavior problems (which include repetitive behaviors and SIBs) in ASD adolescents and adults in a longitudinal study [335]. Stress and comorbid psychiatric disorders have been found to substantially explain an association of autistic traits and repetitive behaviors in ASD [336]. Conversely, the Preschool Autism Communication Trial (PACT) intervention that decreased aspects of ASD symptomatology was found in a follow-up study to have led to markedly reduced rates of repetitive behaviors, suggesting the latter are at least partly sequelae of ASD symptomatology [34]. The availability of praise, emotional warmth, and high relationship quality, which likely ameliorate dysphoric emotions, have been found to reduce repetitive behaviors in ASD adolescents and adults [337]. In sum, heightened dysphoric emotions are elicited by numerous ASD features, and relief of them is likely an important motivator of stereotyped behaviors and SIBs.

### 5.4. Additional Brain Regions Display Likely Secondary Structural and Functional Atypicalities in ASD

Additional brain regions display structural and functional abnormalities in ASD, predominantly visual cortical areas, inferior frontal gyrus (IFG) and other regions of the PFC, caudate nucleus and putamen in the basal ganglia, hippocampus, sensorimotor cortex, cerebellum, and thalamus [45,46,47,48,49,338]. It is suggested that some of these abnormalities are sequelae rather than etiological disruptions.

#### 5.4.1. Visual cortex 

See Figure 1, Box 7.

Visual cortical areas processing motion and ventral temporal cortex (VTC) subregions have been found by structural neuroimaging to be abnormal in ASD [45,190]. In addition, functional neuroimaging meta-analyses have reported MTG and FG hypoactivity in social tasks in ASD versus TD controls [46,88]. 

At the neural level, as previously summarized, the amygdala relays strongly and reciprocally with the visual cortex, and these regions engage in recurrent processing [182,217,225,226,227,228]. Conversely, amygdala dysfunction caused by surgical lesions in monkey or Urbach–Wiethe or other diseases in human induces hypoactivation of the visual cortex to emotional stimuli, as well as structural degeneration of it relative to healthy controls [201,231,339,340]. At the behavioral level, social network size in monkeys and in humans has been found to be positively associated with VTC and STS volumes [341,342]. Moreover, as the monkeys were independently allocated to different sized groups, variations in network size likely drove the variation in visual cortex volumes [342]. 

In ASD, visual cortex structure and activity are likely diminished by enduring amygdala dysfunction and by the reduced social network size reported in this disorder [146]. Thus, visual cortex anomalies are likely driven at least partly by neural and behavioral features of ASD, but the extent of their contribution remains to be quantified.

#### 5.4.2. IFG 

See Figure 1, Box 8.

There is some evidence of IFG disruption in ASD. Grey matter volume was found to be reduced in IFG in ASD versus TD controls [190], and a meta-analysis of functional neuroimaging investigations of social tasks reported hypoactivation of IFG (BA 44) in ASD compared to TD controls [46]. 

The heavy structural interconnections of subregions of IFG with the amygdala, OFC, temporal regions, and insula [223,278,343], which are commonly disrupted in ASD, likely lead to dysfunction of IFG. IFG participates in the representation and execution of expressive behaviors, as well as in syntax and in high-level cognitive processes [277,281,344,345,346,347]. Thus, IFG dysfunctions likely participate in the ASD symptoms of impaired expressive behaviors and gestures [58], the frequent impairment of syntax reported in verbal individuals [9], and the difficulties in goal-directed and planning aspects of cognition [168]. In sum, dysfunctions of IFG are hypothesized to be substantially driven by disruption of the social brain regions, and to manifest in conspicuous features of ASD which are thus probably sequelae.

#### 5.4.3. Caudate Nucleus 

See Figure 1, Box 9.

Caudate nucleus enlargement in ASD relative to TD controls has been reported by several meta-analyses of structural neuroimaging studies [45,47], and by a cellular study that examined this region [49]. An age-related finding was also reported: ASD caudate nucleus volume in childhood is reduced or the same as that of TD controls, but in adolescence and adulthood is significantly enlarged [45,49]. 

The basal ganglia, particularly the caudate nucleus, are the predominant regions in the mediation of stereotypies and repetitive behaviors [348,349,350]. It is hypothesized that these behaviors are motivated to relieve dysphoric emotions (see Section 5.3), and that their repeated performance drives enlargement of the brain region that implements them. Consistent with these hypotheses, significant caudate nucleus enlargement emerges from later childhood onwards, and the level of repetitive behaviors has been found to correlate with caudate nucleus volume or growth [45,49,351,352,353,354]. Nevertheless, some null or inconsistent findings have also been reported (see for review, [355]). Overall, caudate nucleus enlargement is at least partly a sequela of ASD, but some inconsistent findings suggest further mechanisms may be involved. 

#### 5.4.4. Hippocampus 

See Figure 1, Box 10.

Hippocampal abnormalities of functional hypoactivation and of reduced grey matter volume in ASD relative to TD controls have been reported by meta-analyses of functional and structural neuroimaging studies [45,46,48]; (but see [47] for contrary structural findings). 

Factors that can impair hippocampal structure and function include persistently elevated cortisol levels (see for reviews, [356,357]), epilepsy [358,359,360], and lesions of the vestibular system [361]. In addition, neonatal amygdala lesions in monkeys non-significantly reduced hippocampal volume in these animals [339]. Poor sleep quality disrupts the consolidation processes that the hippocampus implements during sleep, thereby impairing hippocampal function [254]. All these factors commonly operate in ASD. Cortisol levels are persistently elevated, as measured by hair cortisol, and impair hippocampal function as assessed by a spatial working memory task [151]. The prevalence of epilepsy in ASD is 10–20% [172], vestibular impairments are reported [19,63,114,115], the amygdala is disrupted (see Section 3.1), and poor sleep quality is frequent [173]. Thus, hippocampal anomalies are likely substantially driven by features of ASD, but the extent of their contribution remains to be quantified.

### 5.5. Heterogeneity 

Heterogeneity is a widespread feature of disease [362], but it is particularly marked in ASD, and a number of the latter’s features may contribute to this. Social brain regions are richly multifunctional; for instance, a rich diversity of behavioral and cognitive functions is mediated by the amygdala [219]. Correspondingly at the neural level, the amygdala is unusually widely interconnected; it interconnects with regions whose processing ranges from visual objects, to features of foods in the mouth, pain, pleasure, blood acidity, stress hormones, high-level cognition, spatial knowledge, episodic memory, and conceptual and expressive motor functions [182,223,363,364,365,366]. Different numbers, combinations, and extents of dysfunctions of a richly multifunctional brain region will produce great diversity of symptoms and features. Further, any specific dysfunction is unlikely to be found universally in ASD [181].

ASD is a developmental disorder, and insults sustained during versus after critical periods of a neural circuit’s development produce markedly different impairments. The visual system illustrates this: V1 lesions sustained early in development induce plasticity in a parallel pathway and its target V5, resulting in little visual impairment. V1 lesions sustained later in development cannot induce such plasticity as the parallel pathway atrophies, resulting in substantial visual impairment [367]. Also, ASD is an unusually complex disorder, and variation in interactions among systems may manifest as heterogeneity. For instance, intact OFC appears capable of compensating for amygdala lesions [200,201] so can reduce impairments, but lesions of OFC itself diminish this capability and increase impairments. Again, multiple ASD features (enduring cortisol elevation, epilepsy, and vestibular deficits) affect hippocampal structure and function, so variation in these features will manifest as heterogeneity of ASD hippocampal abnormalities. In sum, the heterogeneity and inconsistency of ASD symptomatology is particularly severe, in part because ASD is a developmental disorder, involves brain regions with unusually diverse functions at a fine-grained level, and is unusually complex giving rise to a multiplicity of interactions. 

## 6. Summary, Cautions, and Causation

### 6.1. Summary 

The research question examined in this work is: What are the disordered neural circuits that explain ASD symptomatology in all its richness? The answer proposed is the hypothesis that four social brain regions, the amygdala, OFC, TPC, and insula, are commonly disrupted in ASD and largely constitute the pathogenic mechanism of the disorder. These neural disruptions and resulting ASD symptoms are also drivers of diverse secondary features and of anomalies of several additional brain regions that together contribute a substantial proportion of overall ASD symptomatology. Together, the model explains a high proportion of ASD symptoms and features, is consistent with findings that ASD is essentially a unitary disorder [36,37,38], and it relates symptoms and features to neurocircuitry disruptions in accordance with the RDoC framework [54,55,56].

### 6.2. Cautions 

Some experimental findings appear to question amygdala involvement in ASD. Two adult female patients with amygdala lesions due to Urbach–Wiethe disease were comprehensively examined, but the findings were that they did not come close to meeting criteria for ASD, nor any other psychiatric disorder [368]. Notwithstanding, one of these subjects was earlier tested with the Heider and Simmel [125] paradigm, and was found to perform similarly to ASD subjects rather than TD subjects [369]. Overall, further fine-grained abnormality-focused testing of hypothesized amygdala-related abnormalities is warranted. Conversely, involvement in ASD of brain regions beyond the amygdala is supported. 

A further major issue is that the four social brain regions are likely linked by properties or vulnerabilities in common. For instance, all are high-level, heavily multisensory, structured as hubs, or may have further commonalities yet to be elaborated. Many ASD symptoms are likely multifactorial, but only hypothesized major factors have been set out. For example, sleep problems in ASD are promoted by such factors as stress, anxiety, depression, epilepsy, gastrointestinal disorders, hyperactivity, OFC dysfunction, and medications [156,170,171,173], although only major factors are considered. At the neural level, distributed neural networks typically perform a function, so dysfunctions of several regions in a network may be reported to participate in a specific symptom. 

The involvement in ASD of several further brain regions is unresolved. Significantly increased neuron numbers in PFC in infancy have been reported by a cellular study [338]. The prevalence and causes of this abnormality and any contribution to ASD symptomatology are currently unknown. Somatosensory cortex anomalies have been reported [45], but the causes and the contributions to symptomatology have been little studied. A number of sensory systems relay through this region [258], so the well-established sensory impairments of ASD may to an unknown extent explain these anomalies. 

Structural and functional abnormalities of the cerebellum and its vermis subregion have been reported in ASD [45,46,47,49,370,371]. Findings on the cerebellum, however, can be affected by diverse factors. Lesions of TPC and hippocampus affect the cerebellum, as do age, IQ level, total brain volume, anticonvulsant medications, methodological issues, and perhaps other factors [47,49,289,360,371,372]. Functions of the cerebellum include substantial contributions to posture and gait, limb coordination, components of speech, and oculomotor coordination [373]. Cerebellar dysfunction has thus been suggested to contribute to ASD impairments in posture and gait [119]. Overall, the nature and etiology of cerebellar anomalies in ASD require further investigation. A structural meta-analysis has reported modest abnormalities of the thalamus [47], and a study of metabolites in ASD and TD twin pairs reported several significant abnormalities in ASD [374]. Further structural meta-analyses and a cellular study that examined it have not reported thalamic abnormalities [45,48,49]. Together, the involvement of thalamic abnormalities in ASD requires further study.

### 6.3. Causation 

The causation of ASD is poorly understood, but a number of risk factors have been identified. The contribution of genetic factors to variance in ASD was previously thought to be approximately 90%, but larger and more recent studies have suggested a figure closer to 50% [375]. Much remains to be learned about the probably hundreds of genes that are likely involved [376]. A number of environmental risk factors have been reported, and are of great interest for the possibility of lowering ASD incidence. Birth complications are a major group of risk factors, and involve such strong factors as birth injury, meconium aspiration, hypoxia, maternal hemorrhage, or poor neonate state as indexed by a low Apgar score [2]. Low birth weight of the neonate is a further strong risk factor [2]. Maternal condition-related risk factors include migration during gestation, vitamin D and other nutritional deficiencies, as well as advanced age of either parent [2]. Toxicants such as pesticides and air pollutants are further risk factors [2,3,4]. Thus, current knowledge suggests diverse risk factors increase the likelihood of ASD, but how they contribute to the common disorder outcome is unclear. 

## 7. Testable Predictions and Directions for Future Research

The four social brain regions are predicted to be impaired to varying degrees in ASD, independent of the causal mechanism involved. Their integrity may be indexed by stimulation of multiple sensory systems and neuroimaging, which are predicted to reveal some abnormal or blunted responses in the four social brain regions, relative to those in TD controls. Diverse sensory deficits in ASD are reported in the literature, but evidence is unsystematic and mostly gained through questionnaire methodologies, which are unsatisfactory [110,377]. Systematic studies using objective, quantitative, and standardized paradigms are needed to characterize comprehensively and quantitatively the sensory system deficits in ASD individuals [110]. Further, studies of sensory system disruptions may cast light on the causation of the amygdala cellular abnormalities that chiefly affect the lateral amygdaloid nucleus, which receives the heaviest sensory inputs [49,182,183]. 

Intangible knowledge is hypothesized to be markedly impaired in ASD. This may be tested with established cognitive paradigms, particularly the property-listing task and its variants. These involve subjects listing all the features they can think of for each presented stimulus, or providing quantitative ratings of the influence of specified features in those stimuli [378,379]. Intangible, top-down features (e.g., valence) that characterize social images are major drivers of TD scanpaths, whereas due to impaired intangible knowledge it is predicted that visual, bottom-up features of social images are main drivers of ASD scanpaths, and this could be tested with eye-tracking paradigms [75,222]. ASD individuals express apparent disinterest in such sounds as their own name and human language, and this may be a manifestation of failure to enhance neural responses to significant stimuli. Such failure to enhance activation of visual or auditory cortices is hypothesized to be driven by amygdala hypoactivation, and this could be tested by neuroimaging studies of activations and effective connectivity during processing of such stimuli.

OFC disruption is hypothesized to participate in multiple ASD features. Empirical testing of these hypotheses may be performed with neuroimaging optimized for OFC, which is otherwise difficult to neuroimage adequately [197,198,199], combined with established measures and paradigms. These include the menu paradigm [245,380] for assessing food representations; the Neuropsychiatric Inventory [381] for anger, aggression, and inappropriate behaviors; the physical activity monitor (actigraph), polysomnography, and Bedtime problems, Excessive sleepiness, Awakenings, Regularity of sleep, and Snoring (BEARS) questionnaire [173,382] for sleep disruption; and the Cambridge Gamble Task and the Iowa Gambling Task [197,237] for decision-making and planning. 

TPC disruption is hypothesized to participate in multiple ASD features. There is some evidence of impaired body representation in ASD but little formal study. Such deficits may be examined by requiring ASD individuals to point to various body parts, or to estimate their sizes, as well as semi-structured interview covering body knowledge, difficulties in dressing, and other body-related functions. Such findings may also provide insights into aspects of daily living impairments, which may be measured with the Waisman Activities of Daily Living Scale [383]. 

The strength in concrete-level cognition in ASD and its hypothesized associations with weakness in intangible cognition, and with impaired conceptual processing and generalization, may be tested with established measures and their correlations. These include the property-listing paradigm and its variants [378,379], the sorting paradigm of Ropar and Peebles [135], the Halstead Category Test, the Trail Making Test–Part B, and the Verbal Concept Formation Task [304,384,385]. Conceptual processing may also be negatively associated with measures of insistence on sameness, and rigid repetitive routines and rituals, which may be assessed with the Repetitive Behaviors Scale–Revised [386]. IFG hypoactivation is hypothesized to contribute to impoverished expressive behaviors, and this may be tested by neuroimaging. Hippocampal structural abnormalities are hypothesized to be driven in part by elevated stress hormone levels and vestibular dysfunction, and this could be tested with structural neuroimaging, hair cortisol measures, and vestibular tests. The role of stress and dysphoria in evoking stereotypies and SIBs may be explored with wearable telemetric devices that measure and transmit heart rate, blood pressure, respiration, motor activity, and other data [323,387]. These data may be related to observations of precipitating events, execution, and physiological effects of stereotypies and SIBs. Corresponding variables for possible ameliorative events such as receiving praise or comfort also merit investigation. 

The elucidation of biomarkers to facilitate early, rapid, and objective diagnosis of ASD is an urgent challenge. A biomarker is defined as “a biological feature that can be objectively measured and that serves as an indicator of normal or pathogenic biological processes” [388] (p. 1753). Given the heterogeneity of ASD, multiple biomarkers that possess low sensitivity but high accuracy and specificity may be a realistic target [142]. A number of such tests suggested above if validated, or already suggested by researchers, may be valuable, and may include: abnormal responses to stimulation of the four social brain regions; impaired intangible knowledge; abnormal eye gaze patterns [76,389]; preference for physical or concrete stimuli [135,141,142]; impaired body knowledge and interoceptive functions (for definition, see [294,390]; [112,114,115,122]); motor impairments [21,121]; enduring stress expressed in elevated hair cortisol [151]; motor stereotypies and repetitive behaviors [17,391]; and enlarged head circumference and brain volume in early childhood [174]. A novel proposal derives from findings that multiple brain regions are structurally abnormal in ASD, mostly involving grey matter loss. These abnormalities may commonly involve neurodegenerative processes, which are associated with biomarkers such as focal iron deposition (cf. [51,52]). Thus, patterns of iron deposition distributed across brain regions implicated in ASD, may provide an early quantitative biomarker for ASD that can be measured by neuroimaging. In sum, the presented model offers testable predictions about the neurocircuitry disruptions that participate in specific ASD symptoms and features, and about interrelationships among symptoms. These predictions can be tested with established experimental paradigms and neuroimaging.

## 8. Conclusions

A rich and valuable body of findings is accumulating on ASD. The presented model of ASD builds on these findings and hypotheses, to offer a provisional account of the pathogenic brain regions that underlie this disorder; these are the amygdala, OFC, TPC, and insula. Further brain regions are affected secondarily. The model also offers an extensive and detailed etiological account of ASD symptomatology, is consistent with a unitary view of ASD, and with the heterogeneity of it. The model makes detailed testable predictions that should further illuminate the mechanisms of ASD symptoms. Biomarkers for more rapid diagnosis are summarized, and a novel one proposed. Together, the model should facilitate further theoretical progress, and foster the development of personalized, more efficacious interventions.

## Figures and Tables

**Figure 1 brainsci-09-00130-f001:**
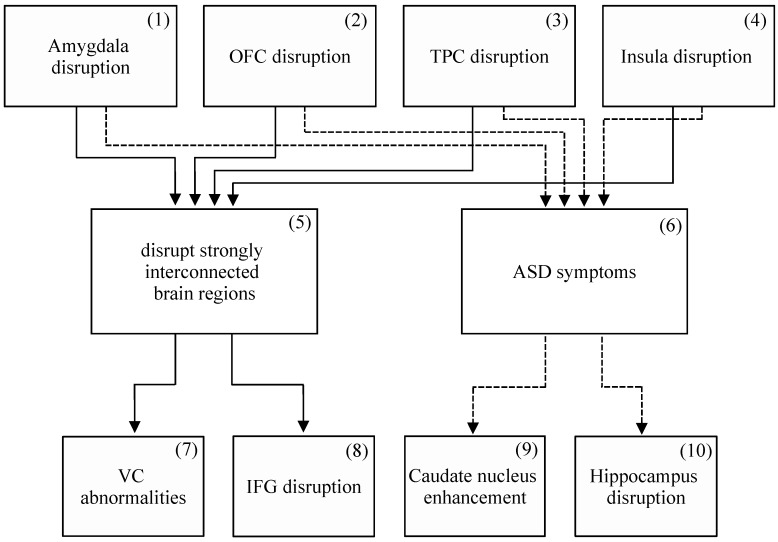
Summary of disrupted neurocircuitry. Four social brain regions are commonly disrupted and these disruptions and the resulting symptoms drive additional abnormalities of the visual cortex, inferior frontal gyrus, caudate nucleus, and hippocampus. ASD, autism spectrum disorders; IFG, inferior frontal gyrus; OFC, orbitofrontal cortex; TPC, temporoparietal cortex; VC, visual cortex.

**Figure 2 brainsci-09-00130-f002:**
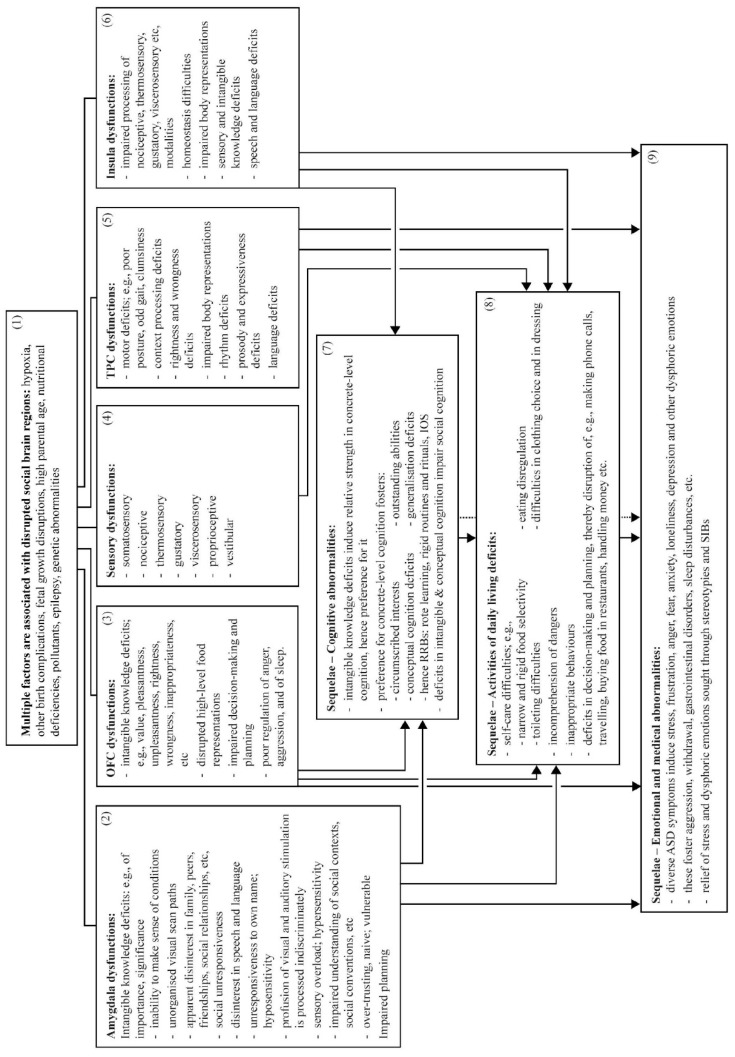
Disruption of four social brain regions extensively explains ASD symptomatology. Multiple factors are causally associated with ASD, and presumably disrupt assembly of social brain regions and circuits. ASD, autism spectrum disorders; IOS, insistence on sameness; OFC, orbitofrontal cortex; RRBs, restricted repetitive behaviors; SIBs, self-injurious behaviors; TPC, temporoparietal cortex.

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
