# Peer review of "Four Social Brain Regions, Their Dysfunctions, and Sequelae, Extensively Explain Autism Spectrum Disorder Symptomatology"

_brainsci, 2019, doi:10.3390/brainsci9060130_

Round 1
Reviewer 1 Report
This is a very interesting and detailed review, which summarizes the current state of the art on the involvement of four major brain areas (amygdala, orbitofrontal cortex, temporoparietal
cortex, and insula) in autism spectrum disorder (ASD) pathogenesis. The author made a great job in summarizing findings from many different types of studies ranging from psychology to brain imaging, and ended up in proposing a model ASD pathogenesis based on the dysfunction of tthe above mentioned brain areas. In my opinion this is a sound research, which might guide future research and (most importantly) clinical studies.
Author Response
Reviewer 1: Thank you very much for the positive comments. No amendments are called for, so none are made.
Reviewer 2 Report
The author summarized and discussed the hypothesis on the mechanisms underlying autism spectrum disorder (ASD) pathology, especially focusing on four brain regions, amygdala, orbitofrontal cortex, temporoparietal cortex, and insula. The manuscript is written well and the information is very useful for the integrated understanding for brain neuronal networks for ASD etiology. However, Figure 1 is complicated and difficult to read and quickly scan the information, because it is composed of only sundry letters.Thus, I recommend you to replace the current Figure 1 by an illustration schema indicating the relations between brain regions simply to make it easier for readers to understand, before next submission.
Author Response
Reviewer 2: Thank you very much for the positive comments. An amendment is recommended, and calls for a diagram that illustrates the relations between brain regions. This is now included as Figure 1, and in effect sets out the neurocircuitry-level disruptions.
It is also recommended that the original Figure 1 is deleted. I am loath to do this, however. The figure matches the organization of the text, and deals with the symptom-level of the model. The figure thus gives a ‘map’ of the text to guide the reader, a preview of symptoms sections, and summarizes inter-connections among symptoms, and among brain regions and symptoms. I must also admit I don’t quite understand the comment that the figure is ‘composed of only sundry letters’. Together, I think the figure is of value to readers, and that its advantages outweigh its disadvantages, so have retained it as Figure 2, which I hope is acceptable.